# Emergence of sliding ferroelectricity in naturally parallel-stacked multilayer ReSe$_2$ semiconductor

Wuhong Xue [1,2,5], Peng Wang[1,2,5], Wenjuan Ci[1,2,5], Ying Guo[3], Jingyuan Qu[1,2], Zeting Zeng[1,2], Tianqi Liu[1,2], Ri He [4] ✉, Shaobo Cheng [3] ✉ & Xiaohong Xu [1,2] ✉

Sliding ferroelectric semiconductors can advance the applications of slide-tronics in silicon-compatible microelectronic and optoelectronic devices for the post-Moore era. However, traditional sliding ferroelectrics typically require complex artificial stacking to break symmetry, and most of them lack atomic-scale evidence. Here, we report a sliding ferroelectric semiconductor of naturally parallel-stacked ReSe$_2$ with layer number $N \geq 3$ through rational sliding approach, which is different from the reported sliding ferroelectrics with $N \geq 2$. This sliding ferroelectricity avoids the rigorous artificially stacking design. The origin of ferroelectricity arises from the fact that the sliding of arbitrary middle atomic layer breaks the spatial inversion symmetry. We also directly measure the polarization value of ReSe$_2$ by polarization-electric field hysteresis. Additionally, the electro-tunable ferroelectric polarization is further confirmed by microscopic ferroelectric switching. And, the sliding ferroelectricity enables switchable ferroelectric diode and programmable photovoltaic effect. Our study sheds light on the exploration of sliding ferroelectric semiconductors in naturally parallel-stacked configurations.

Sliding ferroelectricity often occurs in the manually stacked bilayer configurations of two-dimensional (2D) materials. The out-of-plane polarization is flipped by in-plane sliding one layer by a fraction of the unit cell[1–3]. This endows sliding ferroelectrics with many advantages, including not requiring polar crystals[4], ultrafast and fatigue-free switching[5,6], and low polarization reversal barrier[6]. Initially, Wu et al. proposed the sliding ferroelectric mechanism and predicted that BN and MoS$_2$ can generate out-of-plane polarization by sliding engineering[7]. This theoretical prediction was subsequently verified through extensive experiments, such as topological semimetal WTe$_2$[4,8], insulator BN[1,2], semiconductors (MoS$_2$, WSe$_2$, WS$_2$, InSe)[9–15], and single crystal[16]. These pioneering progresses open up a research field called slidetronics[1,2]. Noteworthy, the sliding ferroelectric semiconductors with moderate band gap and the efficient gate tunability exhibit significant application potentials in multifunctional information storage[12,17,18], ferroelectric photovoltaics[13,15,19], and neuromorphic computing[15,20]. Therefore, they have great potential to advance the development of slidetronics.

As a typical sliding ferroelectric semiconductor, 3R-stacked bilayer MoS$_2$ is only $C_3$-symmetry, having neither an inversion center

[1]Key Laboratory of Magnetic Molecules and Magnetic Information Materials of Ministry of Education, School of Chemistry and Materials Science, Shanxi Normal University, Taiyuan, China. [2]Research Institute of Materials Science, Shanxi Key Laboratory of Advanced Magnetic Materials and Devices, Shanxi Normal University, Taiyuan, China. [3]Henan Key Laboratory of Diamond Optoelectronic Materials and Devices, Key Laboratory of Material Physics, Ministry of Education, School of Physics and Microelectronics, Zhengzhou University, Zhengzhou, China. [4]Key Laboratory of Magnetic Materials Devices, Zhejiang Province Key Laboratory of Magnetic Materials and Application Technology, Ningbo Institute of Materials Technology and Engineering, Chinese Academy of Sciences, Ningbo, China. [5]These authors contributed equally: Wuhong Xue, Peng Wang, Wenjuan Ci. ✉e-mail: heri@nimte.ac.cn; chengshaobo@zzu.edu.cn; xuxh@sxnu.edu.cn

nor a mirror symmetry plane. This symmetry breaking allows for charge transfer between adjacent atomic layers, resulting in spontaneous polarization along the out-of-plane direction[10]. However, constructing such systems typically relies on complex manual stacking or specific growth techniques. Notably, an important work has reported the enhanced sliding switchable polarization by modifying the microstructure of InSe system through Y doping strategy[21], which avoids the complex artificial stacking. To further broaden the selection scope of material, it is crucial to explore intrinsic sliding ferroelectric semiconductor systems without artificial stacking and chemical modification. ReX$_2$ (X = S, Se) exhibits a twisted octahedral structure (1T' phase) with different natural stacking configuration (parallel and antiparallel)[22,23]. From the perspective of point groups, monolayer ReX$_2$ belongs to the $C_i$ point group and only contains one symmetry operation[24], namely inversion of the central symmetry, which does not produce out-of-plane polarization. For ReX$_2$ of antiparallel stacking with layer number $N \geq 2$, the out-of-plane polarization was observed, and the electric polarization can be switched by interlayer sliding[25,26]. However, in parallel-stacked ReX$_2$ systems, whether vertical polarization can be induced by sliding engineering remains unexplored, such studies would be useful in understanding the physical origin and spurring further research into sliding ferroelectric semiconductor system. Therefore, it is crucial to explore in depth the sliding approach for realizing ferroelectricity in naturally parallel-stacked multilayer ReX$_2$ systems and to elucidate the mechanism on the atomic scale.

Here, we demonstrate sliding ferroelectricity in parallel-stacked ReSe$_2$ semiconductor with layer number $N \geq 3$, which is different from the conventional stacking structure. The first-principles calculations uncover the origin of ferroelectricity that the sliding of arbitrary middle atomic layer breaks the spatial inversion symmetry, and the displacements of microscopic atomic configuration are directly visualized using scanning transmission electron microscopy (STEM). Meanwhile, piezoresponse force microscopy (PFM) measurements demonstrate the visualized ferroelectric domains and ferroelectric switching property. As the direct observation of sliding ferroelectricity, the ferroelectric hysteresis loop shows an experimental polarization value of -0.67 μC/cm². In addition, the switchable ferroelectric diode effect and the programmable ferroelectric photovoltaic effect are successfully demonstrated in the ReSe$_2$ based sandwiched device, further confirming sliding ferroelectricity. This work avoids the rigorous artificially stacking design required for conventional sliding ferroelectrics and expands the family of sliding ferroelectric semiconductors into the naturally parallel-stacked multilayer ReX$_2$.

## Results

ReSe$_2$ is a high ambient stable van der Waals material where each layer is composed of a Re atoms plane sandwiched between two Se atoms planes[27]. Different from the most common group-VI transition metal dichalcogenides (e.g., MoS$_2$ and WS$_2$), there are unbound valence electrons outside the nucleus of the Re atom in ReSe$_2$. Due to the Jahn-Teller effect (or Peierls distortion), the adjacent Re atoms are bonded in the form of zigzag Re4 clusters and align along b-axis direction to form Re4 chains, forming a distorted octahedral (1T') structure and reducing the symmetry of structures (Fig. 1a)[28–30]. In addition, the Peierls distortion greatly weakens the van der Walls interaction in the interlayer[31]. The low-symmetry structure and weak van der Waals interlayer interactions of ReSe$_2$ endow it the unique ability of interlayer sliding[32], which facilitates the realization of low-potential-barrier interlayer sliding. ReSe$_2$ crystal was synthesized using the chemical vapor transport method, and ReSe$_2$ flakes of different thicknesses were mechanically exfoliated using polydimethylsiloxane (PDMS) films to further verify their atomic structure and physical properties (More

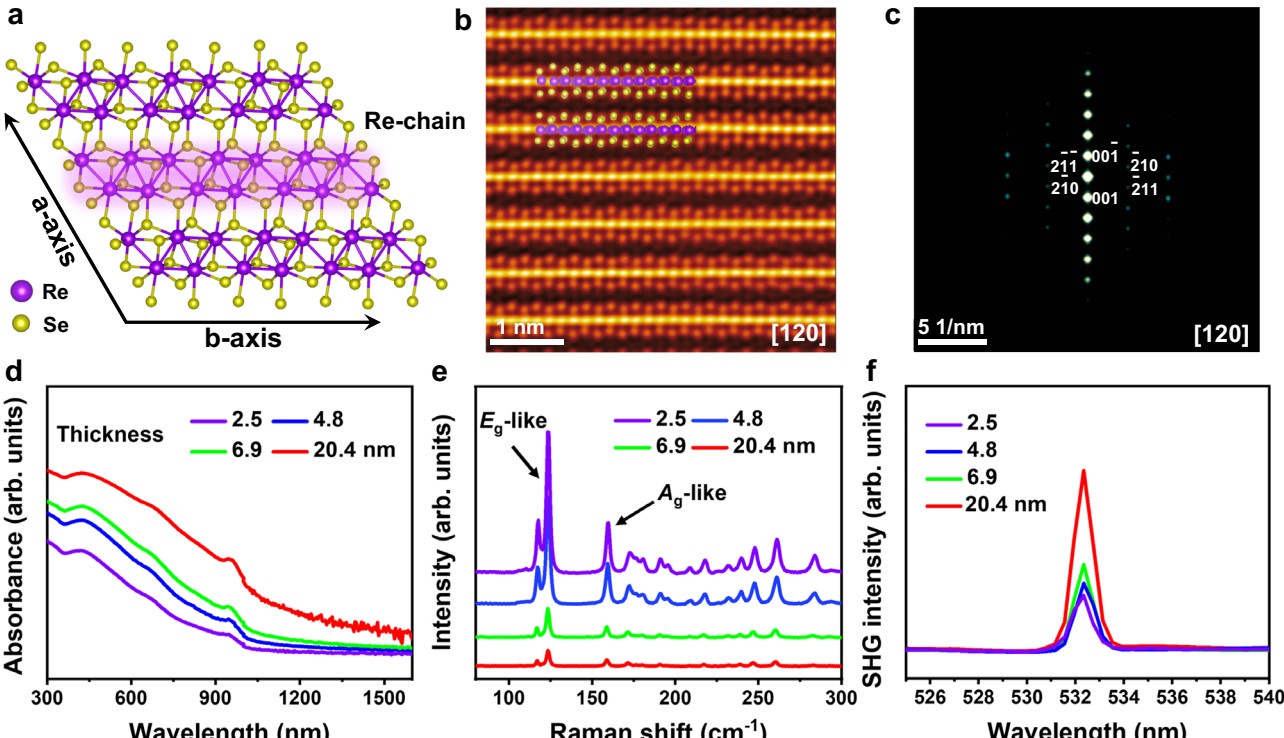

**Fig. 1 | Crystal structures and spectral characteristics of ReSe$_2$ flakes. a** Top-view of the atomic arrangements in monolayer ReSe$_2$. The four covalently bonded Re atoms in the unit cell arrange in a parallelogram and the adjacent parallelograms connect by covalent bonding along the b-axis forming a Re-chain. Purple and yellow spheres represent Re and Se atoms, respectively. **b, c** HAADF image of mechanically exfoliated ReSe$_2$ flake along the [120] direction and the corresponding SAED pattern. UV-Vis-NIR absorption spectra (**d**), Raman spectra (**e**), and SHG signals (**f**) of ReSe$_2$ with different thicknesses.

details can be found in "Methods"). Figure 1b, c shows atomic-level high-angle annular dark-field (HAADF) image and selected area electron diffraction (SAED) pattern of ReSe₂ flake along the [120] zone axis. The results exhibit high-quality layered parallel stacking structure. The ultraviolet-visible-near infrared (UV-Vis-NIR) absorption spectra of samples with different thicknesses show the sharp absorption edge near 950 nm (Fig. 1d). From the Tauc plot, the band gap was calculated to be ~1.1–1.2 eV, which is very close to the reported value[27,33]. The transfer characteristic curve indicates that ReSe₂ is an ambipolar semiconductor material (Supplementary Fig. 1). From Raman results of ReSe₂ with different thicknesses (Fig. 1e and Supplementary Fig. 2), the peaks at 124 cm⁻¹ and 173 cm⁻¹ correspond to the in-plane $E_g$-like mode and the out-of-plane $A_g$-like mode, respectively, and the around 13 modes were observed between 100 cm⁻¹ and 300 cm⁻¹, which is much more than that of other transition metal dichalcogenides[34]. These Raman peaks are mainly attributed to the low crystal symmetry of ReSe₂ and related to fundamental in-plane and out-of-plane Raman modes[27]. The anisotropy of ReSe₂ flake, caused by its low crystal symmetry, was confirmed by the excitation angle-resolved Raman spectra (Supplementary Fig. 3a–d). Subsequently, second harmonic generation (SHG) technique, a reliable tool for demonstrating the non-centrosymmetric structure[35,36], was adopted to characterize the structural symmetry of ReSe₂ flakes. The schematic diagram of the SHG was shown in Supplementary Fig. 4a. Figure 1f illustrates the SHG intensity of samples with different thicknesses. A significant SHG signal at 532 nm was observed under the excitation wavelength of 1064 nm, revealing ReSe₂ flakes have a non-centrosymmetric structure. The power-dependent SHG intensity of ReSe₂ flake (~2.5 nm) was investigated (Supplementary Fig. 4b), and the SHG intensity increased monotonically with the power intensity. We further investigated the effect of crystal structure on signal intensity by polarization-resolved SHG. A clear four-lobed pattern was observed (Supplementary Fig. 4c), showing the variation of SHG intensity with azimuthal angle. The above results confirm that ReSe₂ is a parallel-stacked semiconductor material with a non-centrosymmetric structure.

Next, we studied theoretically using first-principles calculations whether ferroelectric polarization can be induced through sliding mechanism in naturally parallel-stacked ReSe₂. Since ReSe₂ exhibits a distorted octahedral structure (1T' phase), the monolayer belongs to the $C_i$ point group with central inversion symmetry and no vertical polarization occurs. For parallel-stacked bilayer ReSe₂ crystal, a 1 × 1 × 2 supercell containing two layers is insufficient to induce polarization by interlayer sliding because of its inversion symmetry preserve (Supplementary Fig. 5). Therefore, a large 1 × 1 × 6 supercell is needed as shown in Fig. 2a. Starting from the large supercell of parallel stacking ReSe₂ with inversion symmetry, sliding the middle two layers of six layers can break its inversion symmetry. There are two low symmetry stacking modes (state 1 and state 2) appear when the sliding of middle two layers, corresponding to geometric structures in Fig. 2b, c. The atomic arrangement of the intermediate state was shown in Supplementary Fig. 6. The calculation of polarization based on Berry-phase method confirmed the symmetry breaking of state 1 and state 2 stacked configurations. The state 1 and state 2 structures have spontaneous vertical polarization with opposite polarization directions, and the polarization values are calculated to be 4.37 and −4.37 μC/cm², respectively. Therefore, the polarization of ReSe₂ can be switched from upward to downward when the middle two layers sliding, and the first-principles density functional theory calculated transition barrier is ~12.8 meV/f.u. (Fig. 2d). In order to reveal the origin of the out-of-plane polarization, we calculated the charge density difference between the top, middle, and bottom layers in state 1 stacking configuration. As shown in Fig. 2e, it is found that an interlayer net charge transfers after sliding transition, resulting in the accumulation and depletion of electrons in the van der Waals gap.

Furthermore, the influence of layer numbers of ReSe₂ and sliding layer numbers on polarization were studied. The ferroelectric polarization occurs in the parallel-stacked multilayer ($N \geq 3$) ReSe₂ (Fig. 3a and Supplementary Fig. 7). Figure 3b shows the ferroelectric polarization values of sliding different layers in a fixed six-layer system. It can be found that no matter how many middle layers undergo sliding,

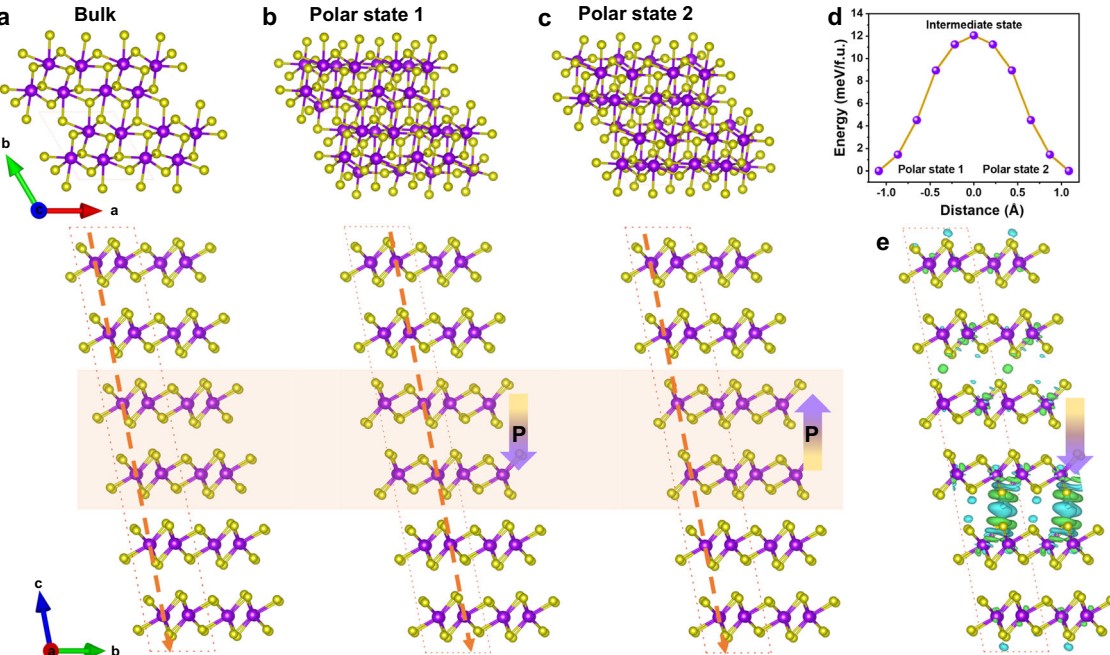

**Fig. 2 | Theoretical analysis of ferroelectricity in ReSe₂ flakes. a** Schematic illustration of the atomic arrangement for parallel stacking. **b, c** Schematic illustration of the atomic arrangement for different polarization directions (state 1 and 2), and the state 2 can be obtained by mirroring the state 1 with respect to the central horizontal plane. **d** Ferroelectric switching pathway for multilayer ReSe₂ from polar state 1 to polar state 2. **e** The charge density difference between the layers in polar state 1 produces a vertical polarization. The arrows indicate the polarization direction. Purple and yellow spheres represent Re and Se atoms, respectively.

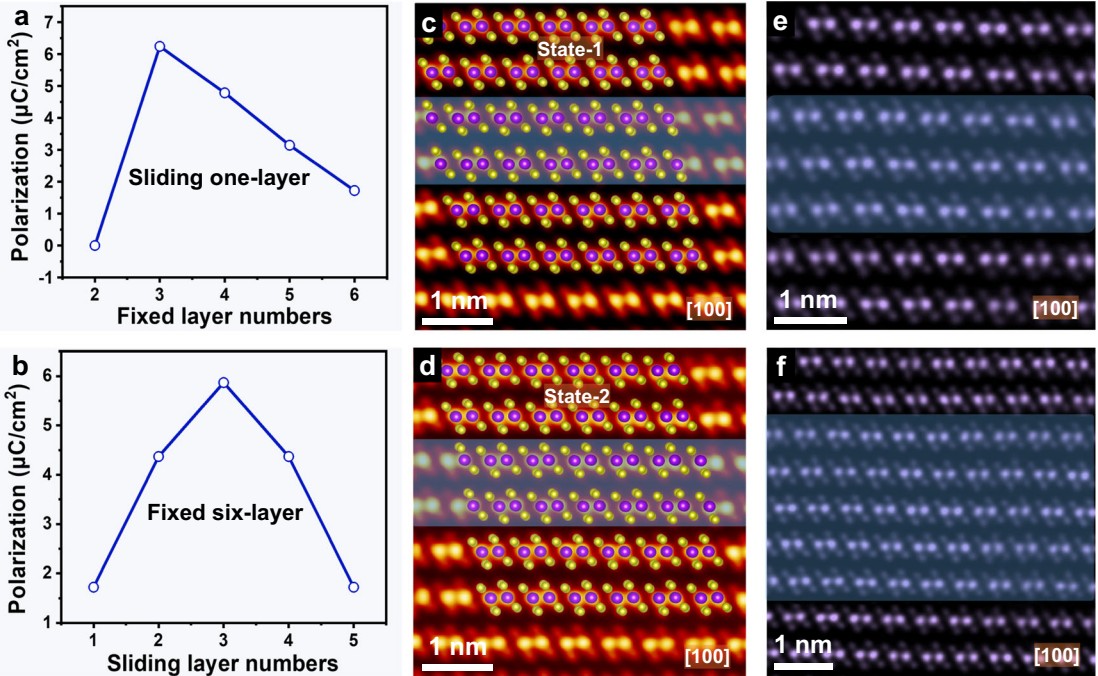

**Fig. 3 | Theoretical and STEM-HAADF microscopic analyses of ReSe₂ sliding ferroelectricity. a** Ferroelectric polarization values of sliding one layer in different layer systems. **b** Ferroelectric polarization values of sliding different layers in a fixed six-layer system. **c, d** HAADF images recorded along the [100] direction for the ferroelectric polarization state 1 and state 2. In the ReSe₂ structural model, purple and yellow spheres represent Re and Se atomic columns, respectively. **e, f** HAADF images of sliding the middle three layers and five layers.

polarization is always generated. These theoretical results indicate that sliding ferroelectricity can be easily obtained through sliding mechanism in naturally parallel-stacked ReSe₂. Excitedly, the sliding of middle layers of ReSe₂ flake was clearly observed from high-resolution STEM-HAADF images. The relative sliding of the middle two layers with left sliding (Fig. 3c) and right sliding (Fig. 3d), corresponding to two opposite ferroelectric polarization states. Atomic configurations of the two polarization states, i.e., state 1 (downward polarization) and state 2 (upward polarization) were placed on top of the HAADF images of Fig. 3c, d. Additionally, we also observed the sliding of the middle three layers (Fig. 3e) and five layers (Fig. 3f). The sliding layers were marked by a shaded box.

Based on the above theoretical prediction of sliding ferroelectricity in parallel-stacked ReSe₂ and direct observation of sliding layers, we experimentally investigated the ferroelectricity of ReSe₂ by PFM technique. PFM is a powerful tool for demonstrating the switchable ferroelectric polarization, enabling non-destructive visualization and modulation of ferroelectric domains[37–41]. The ReSe₂ flakes of different thicknesses used for PFM measurements were obtained through mechanically exfoliation and then transfer onto the conducting Pt substrate (Hefei Kejing Material Technology Co., Ltd supplies) which has good smoothness and conductivity. Figure 4a, d presents the topography images and the height profiles of ~10 L (~6.3 nm) and ~3 L (~1.8 nm) samples. The corresponding out-of-plane PFM phase images at room temperature are shown in Supplementary Fig. 8a, b, revealing a clear phase contrast with multidomain structure. Figure 4b, e shows the PFM phase images of 10 L, 3 L samples after writing box-in-box patterns by reversed DC bias (±4 V), (±3 V). Clear reversal of phase contrast demonstrates the ferroelectric switching in naturally parallel-stacked ReSe₂ flakes. Furthermore, the corresponding local out-of-plane PFM hysteretic loops in phase and amplitude were also recorded. The PFM phase response exhibits a hysteresis loop with a phase contrast of ~180°, while the amplitude response shows a butterfly-shaped curve (Fig. 4c, f), both of which further confirm the robust ferroelectric polarization. However, the ferroelectric polarization was not observed

in bilayer ReSe₂ (Supplementary Fig. 9), which is consistent with the theoretical results. Then, we further evaluated the piezoelectric coefficient $d_{33}$ of an exfoliated ReSe₂ flake with a thickness of ~8.5 nm using PFM. When the AC voltage ($V_{AC}$) is scanned from 1 V to 5 V, the out-of-plane (OOP) amplitude signal of the sample significantly increases (Supplementary Fig. 10). By fitting the amplitude of the sample to the driving $V_{AC}$, the effective $d_{33}$ of the 1T' ReSe₂ flake was determined to be ~3.83 pm V⁻¹. Additionally, the temperature-dependent SHG intensity reveals the Curie temperature of 1T' ReSe₂ to be ~490 K (Supplementary Fig. 11).

Polarization-electric field (P-E) hysteresis loops are regarded as direct evidence for proving ferroelectricity[16,42]. Therefore, we fabricated Au/ReSe₂ (~15.5 nm)/Au vertical device to measure macroscopic ferroelectricity (Supplementary Fig. 12a, d) (more specific preparation details are shown in "Methods" section). The device exhibits the standard P-E hysteresis loops, confirming ferroelectricity of ReSe₂, and the spontaneous polarization ($P_s$) is ~0.67 μC/cm² (Fig. 5a). The discrepancy between experimental and theoretical polarization value may be attributed to the partial switching of the ferroelectric domains and the differences in the layer number of sample and sliding. Additionally, when applying different poling voltages, switchable diode effects are observed (Supplementary Fig. 13), and the rectification ratio reaches ~33 and ~57 with applying poling bias of −5 V and 5 V (Fig. 5b). This can be explained by the single-barrier model of source[43–45], and the positive (negative) bias produces a positive (negative) polarization charge at the source, which further causes a decrease (increase) in the potential barrier height, resulting in an increased (decreased) current (Supplementary Fig. 14). In addition, the nanoscale electrical characterization was performed by Conductive Atomic Force Microscope (C-AFM) and Kelvin Probe Force Microscope (KPFM). An open I-V loop and increasing surface potential appear, which is due to ferroelectric polarization switching (Supplementary Figs. 15, 16). We also constructed BN/Gr/ReSe₂ (~17.9 nm)/BN/Au device to measure the endurance of ferroelectric switches (Supplementary Fig. 12b, e). By applying single pulses with a width of 1 s and amplitudes of $V_{pulse}$ = 3 V

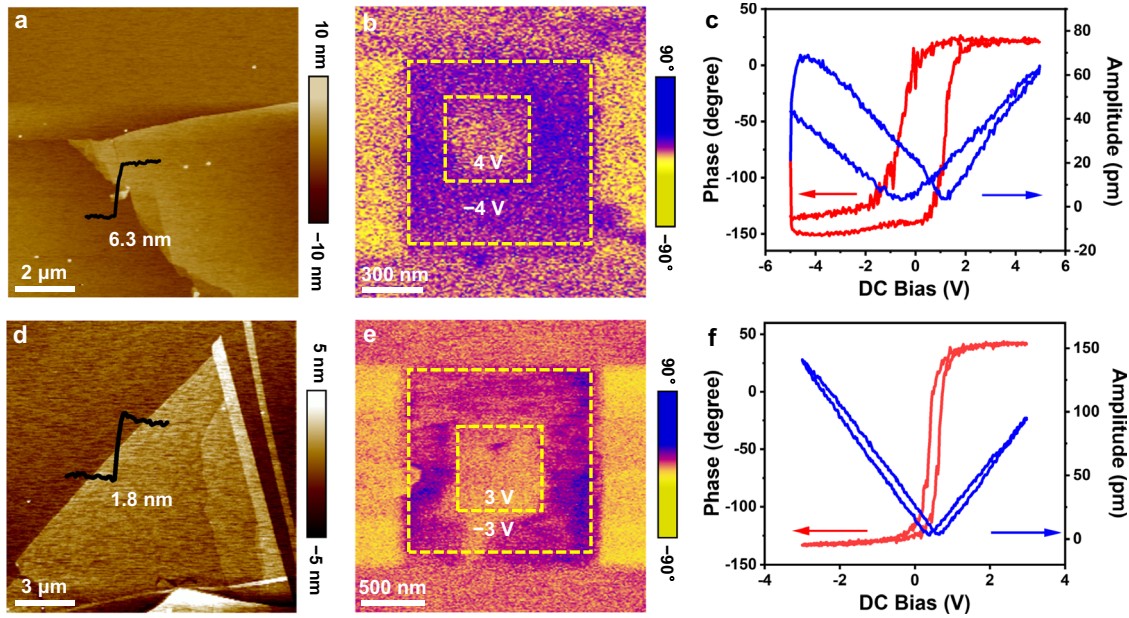

**Fig. 4 | Ferroelectric polarization switching by PFM for multilayer (10 L and 3 L) ReSe₂.** **a, d** Topography, **b, e** Out-of-phase images after writing box-in-box patterns by applying reverse voltage bias. **c, f** Local phase (red) and amplitude (blue) hysteresis loops.

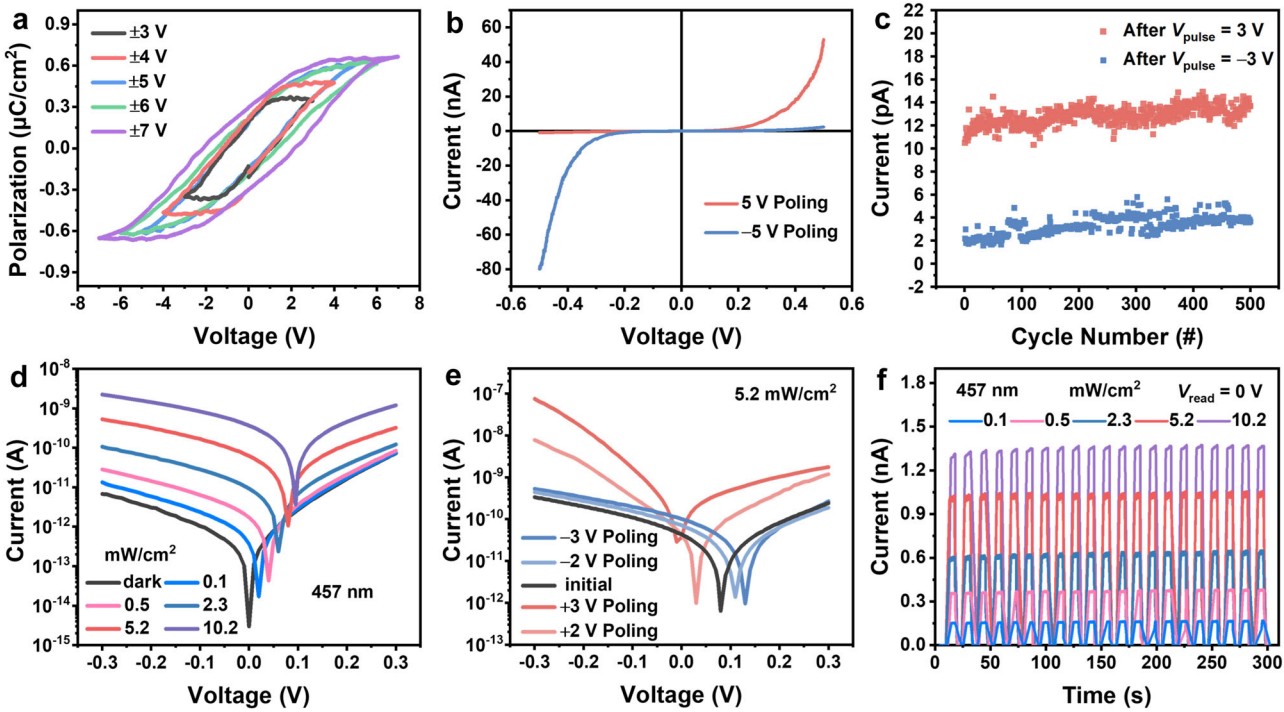

**Fig. 5 | Switchable ferroelectric polarization in ReSe₂ vertical devices.** **a** *P-E* hysteresis loops at different scanning voltages. **b** Switchable ferroelectric diode effect. **c** Endurance of sliding ferroelectrics against switching cycles after applying ±3 V pulse voltages with pulse width of 1 s. **d** *I-V* curves under different light intensities and in the dark. **e** Programmable photovoltaic effect after applying polarization voltages of 0 V (initial), −2 V, −3 V, 2 V, and 3 V, respectively. **f** Time-dependent photocurrent at various light intensities ($V_{read}$ = 0 V).

and $V_{pulse}$ = −3 V repeatedly at $V_{read}$ = 0.5 V, the switching properties of ferroelectric ReSe₂ remain almost unchanged over 500 switching cycles (Fig. 5c). This stable cycle demonstrates the naturally parallel-stacked multilayer ReSe₂ semiconductor has the robustness of polarization switching.

Given the ferroelectricity and semiconductor properties of ReSe₂, it is expected to generate polarization tunable photovoltaics. A Gr/ReSe₂ (~15.8 nm)/Au vertical device was fabricated (Supplementary Fig. 12c, f) to investigate photovoltaic effect through measuring the vertical current (detailed fabrication process described in "Methods" section). The few-layer Gr flake was used as top electrode due to its high optical transmittance in the visible range[13,19]. Figure 5d depicts the *I-V* curves in the dark and under 457 nm illumination with different light intensity from 0.1 mW/cm² to 10.2 mW/cm². As expected, the photovoltaic response of the device is closely related to the light intensity. Both open-circuit voltage ($V_{oc}$) and short-circuit current ($I_{sc}$)

increase with increasing light intensity. This is due to increasing the light intensity induces more electron-hole pairs separated by the built-in electric field, which leads to larger $V_{oc}$ and $I_{sc}$. More interestingly, the photovoltaic effect of the ReSe$_2$ flake is programmable by different polarized electric fields. As shown in Fig. 5e, initially, the device exhibits $V_{oc}$ of 0.08 V and $I_{sc}$ of −43 pA. Compared to widely reported 3R-MoS$_2$, 3R-WS$_2$, ε-InSe, and γ-InSe, larger $V_{oc}$ values were achieved at smaller optical power densities in ReSe$_2$ ferroelectric photovoltaic device (Supplementary Fig. 17). By applying negative poling voltages (−2 V, −3 V), the I-V curves shift rightward and the $V_{oc}$ increases to 0.11 V and 0.13 V, and the $I_{sc}$ decreases to −72 pA and −100 pA. Subsequently, by applying positive poling voltage (+2 V, +3 V), the I-V curves shift leftward and the $V_{oc}$ decreases to 0.03 V and −0.01 V, and the $I_{sc}$ increases to −42 pA and +37 pA. These results demonstrate that the ferroelectric polarization state can enhance or suppress the photovoltaic effect of the ReSe$_2$. This programmable photovoltaic effect is reproducible and it can also be obtained on other ReSe$_2$ device (Supplementary Fig. 18). In addition, we found that the photocurrent can be read out without applying reading voltage, and there are no significant changes in over 20 consecutive switching cycles (Fig. 5f).

## Discussion

In conclusion, we report sliding ferroelectricity in naturally parallel-stacked multilayers ($N \geq 3$) ReSe$_2$ semiconductor. This is very different from reported antiparallel-stacking ReS$_2$ and typical rhombohedral-stacking transition metal dichalcogenides, where the sliding ferroelectricity can appear in bilayer samples. The origin of the sliding ferroelectricity in parallel-stacked ReSe$_2$ is investigated by first-principles calculations, indicating that the intermediate-layer sliding in the multilayer system breaks the spatial inversion symmetry. The noncentrosymmetric configuration was revealed by SHG technology and the intermediate layer sliding is visually observed by STEM. The vertical ferroelectric switching can occur in ReSe$_2$ multilayers ($N \geq 3$), but is not existent in its centrosymmetric bilayers. The P-E hysteresis loops of ReSe$_2$ flake directly show a polarization value of -0.67 μC/cm². In addition, the ReSe$_2$ vertical devices exhibit switchable ferroelectric diode effects and programmable ferroelectric photovoltaic effects. We anticipate that our work will provide a perspective for the research and application of sliding ferroelectrics.

## Methods

### Sample preparation

The bulk single crystal ReSe$_2$, Gr, and BN were purchased from Nanjing MKNANO Tech. Co., Ltd. (www.mukenano.com). The flakes with different thicknesses were mechanically exfoliated from the bulk crystals using polydimethylsiloxane (PDMS). Then, the exfoliated flakes were transferred to the target substrates by the dry transfer technique and heated at 80–90 °C for 10 min to release the adhesion of PDMS to ensure the successful transfer of the flakes.

### Device fabrication

Au/ReSe$_2$/BN/Au devices were prepared by transferring ReSe$_2$ flakes onto SiO$_2$ (300 nm)/Si substrate of pre-deposited Ti/Au (5 nm/40 nm) electrodes by dry transfer technique, and then depositing Au top electrodes on the surface of the flakes. The electrodes were defined using a standard lithography technique and deposited using electron beam evaporation. Additionally, Gr/ReSe$_2$/BN/Au and BN/Gr/ReSe$_2$/BN/Au devices were prepared by sequentially transferring the alternative materials onto the pre-deposited Au electrodes.

### Characterization

The morphology of flakes and devices was characterized by optical microscopy (OM, AOSVI, M330). The Raman spectroscopy (Horiba LabRAM HR Evolution) was used to measure Raman spectra of ReSe$_2$ flakes. Second-harmonic generation (SHG) measurements were conducted using Raman spectrometer equipped with a femtosecond laser (1064 nm) and high-temperature accessories (Linkam THMSG600G). The UV-Vis-NIR spectroscopy (Metatest, MStarter ABS) was used to characterize the optical bandgap. The STEM-HAADF images were acquired by an aberration-corrected JEM-ARM200F microscope operating at 200 kV. During the HAADF image acquisition, the convergence angle was 21 mrad, and the camera length was 8 cm, and the corresponding collection angle ranged from 68 to 280 mrad. The surface morphology and thickness of ReSe$_2$ flakes were measured using tapping-mode atomic force microscopy (AFM, Bruker Dimension Icon). PFM measurements of ReSe$_2$ flakes transferred onto conductive Pt substrates were performed using a commercial atomic force microscope (Bruker Dimension Icon) with an AC resonance mode under the tip-sample contact resonant frequency (~320 kHz). A standardized ferroelectric tester (Precision Multiferroic IcI, Radiant Technologies) was used to measure P-E hysteresis loops. The light sources of different wavelengths were provided by commercially available light-emitting diode lamps, where the power density was measured by FieldMate+PM150X photometer. The electronic and optoelectronic measurements were conducted on Lakeshore's vacuum probe station using a Keithley 4200SCS semiconductor parameter analyzer.

### Density functional theory calculation

The first-principles density functional theory (DFT) calculations were performed by the Vienna ab initio simulation package[46] with projector augmented wave method and the generalized gradient approximation with Perdew–Burke–Ernzerhof exchange-correlation functional[47]. We adopted a plane-wave cutoff energy of 600 eV and a Monkhorst-Pack k-point mesh of $12 \times 12 \times 1$ in the structural relaxation calculations. The optimized lattice constants of parallel-stacked multilayer ReSe$_2$ are $a = 6.63$ Å, $b = 6.75$ Å, and $c = 6.80$ Å with triclinic symmetry. The kinetic of polarization reversal with interlayer sliding pathways were calculated using the climbing nudged elastic band method[48]. The Berry phase method was used to calculate the perpendicular crystalline polarization[49]. The van der Waals interactions in layered ReSe$_2$ are described via the van der Waals correction with optB86b functional in DFT calculations. The spatial charge distribution of the systems was calculated using the Bader Charge analysis[50].

### Reporting summary

Further information on research design is available in the Nature Portfolio Reporting Summary linked to this article.

## Data availability

The dataset of the main figures generated in this study is provided in the Supplementary Information/Source Data file. Source data are provided with this paper.

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

## Acknowledgements

This work was supported by the National Key Research and Development Program of China (Grant no. 2024YFA1410200 (W.X.)), the National Natural Science Foundation of China (Grant nos. U24A6002 (X.X.), 12174237 (X.X.), 12241403 (W.X.), 52371245 (W.X.) and 12204496 (R.H.)), the Shanxi Province Basic Research Program (Grant no. 202303021224009 (W.X.)), and the Shanxi Province Higher Educational Institutions Young Academic Leaders Program (Grant no. 2024Q015 (W.X.)).

## Author contributions

W.X. and X.X. supervised the project. W.X., P.W. and X.X. conceived and designed the experiments. W.X., P.W. and W.C. performed the basic

characterization of the material. P.W. contributed to the device fabrication and characterizations. R.H. performed the theoretical calculation and related discussions. Y.G. and S.C. performed the STEM measurement and related discussions. W.X., P.W., W.C. and X.X. analyzed the results and co-wrote the paper. Z.Z., J.Q., T.L. and other authors discussed the results and commented on the manuscript.

## Competing interests

The authors declare no competing interests.
