## [Transparent Peer Review file · Nature Communications]

Emergence of sliding ferroelectricity in naturally parallel-stacked multilayer ReSe₂ semiconductor

Corresponding Author: Professor Xiaohong Xu

Version 0:

Reviewer comments:

Reviewer #1

(Remarks to the Author)

I have read carefully the work “emergence of sliding ferroelectricity in naturally parallel stacked multilayer ReSe₂ semiconductor”. The paper reports the first observation of sliding ferroelectricity in ReSe₂ semiconductor with parallel stacking providing experimental and theoretical evidence. The paper is relevant as the field of sliding ferroelectricity is of wide interest in the community and the system presented does not require artificial stacking of different 2D materials or at a twisted angle.

The authors first investigate the fundamental physics of the system from a theoretical perspective, then support their findings with direct experimental measurements using multiple well-established techniques. Moreover, they explore the potential integration of sliding ferroelectric ReSe₂ in functional devices

While there are already several Van der Waals systems which present sliding ferroelectricity without artificial stacking e.g. InSe:Y (Ref. 15) or CuInP₂S₆ (ref. 35), the supporting data is solid and I would still recommend the acceptance in Nature Communications with the revisions that I list in the following:

STYLE AND CLARITY

1. Line 80: I would advise against the use of “fantastic” as an adjective for quantifiable properties in scientific writing. I suggest removing it and leave the correct and formal communication “the displacements of microscopic atomic configuration are directly visualized using scanning transmission electron microscopy (STEM)”
2. It would be appreciated a general improvement of the captions, especially the ones of Supplementary that appear approximative and not well deepened and effective.
3. In Fig. 1, panels d-e-f might use the same color for the same thickness to ease readability.
4. I would suggest the authors to improve the legend and color-scale of Fig. S11.

ISSUES

1. Introduction: The authors are right in pointing out that many sliding ferroelectrics based on Van der Waals materials require twisted or artificial stacking to achieve a net switchable polarization. The fact that ReSe₂ does not need artificial stacking is highlighted in the text so it should be the same for the other systems already known for this property. A proper comparison of the advancement with respect to Ref. 15 and Ref. 35 should be given in the introduction.
2. Line 170: “Pt substrate has good smoothness”. Which is the surface roughness? Does it present significant ripples that can favor out of plane polarization? How are these substrates made? There is nothing in the methods section on this.
3. Line 185: for Au/ReSe₂/Au devices thicknesses should be expressed, particularly the number of layers ReSe₂ is fundamental to compare the latter results.
4. Line 190: the difference between theoretical and experimental polarization is attributed to the number of sample and sliding layers: what are possible ratios that would justify this value? Is it possible that the lower value comes from the formation of domains, i.e. a partial switching of the device? If possible, a PFM map taken on a capacitor device would help in clarifying this point.

5. Line 198: Again, a better description of the device stack with layer thicknesses is advised.
6. Line 227: the claim that ReSe2 photovoltaic devices are promising for optoelectronic devices at the current status is far-fetched and unsupported unless some quantitative performance advantage is provided.
7. Fig 5c: the pulse duration employed for the endurance test should be reported in the caption and in the main text

OTHER QUESTIONS:

1. Concerning the power-dependent SHG, such exposition did lead to any degradation of the flake? Did the authors also perform temperature-dependent SHG experiments to assess the Curie temperature of the material?
2. Regarding the PFM measurements, did the authors estimate the piezoelectric coefficient (i.e. piezoelectric modulus d_{33})?
3. Could the authors justify the smaller out-of-plane amplitude of the PFM hysteresis loop with 10 layers with respect to the one with 3 layers (figure 4c-f) ? Did the authors verify how many sliding layers were present in each of the mentioned samples?
4. Could the authors comment on the peculiar left-side of the butterfly hysteresis loop of Fig. 4f? Such branch does not seem to be open as common ones.

Reviewer #2

(Remarks to the Author)

Review on the paper entitled: "Emergence of sliding ferroelectricity in naturally parallel-1 stacked multilayer ReSe2 semiconductor" by W. Xue et al. The authors proposed an alternative way compared to the state-of-the-art to achieve sliding ferroelectricity in two-dimensional ReSe2 systems. By combining both theoretical and experimental results, the authors described clearly the observation of sliding ferroelectricity on multilayers stacked ReSe2, with a number of layers ≥ 3 . The paper is well written and argued. From my point of view, some minor revisions are necessary before publication. My curiosities and suggestions are reported below:

1. A first suggestion concerns the description of material preparation. After the general ReSe2 description in Figure 1, the authors passed to the experimental Raman description without information about the ReSe2 flake preparation, which will be treated subsequently for the final device section. My suggestion is to anticipate the preparation description between the first two figures. This could also help the switching between the theoretical and experimental description during the reading.
2. Have the authors observed some frequency-dependence on the number of ReSe2 layers, like those observed in other 2D TMDs (such as MoS2, WS2...)?
3. Another aspect that was not clarified (or not clear after my readings) is the reason of these sliding on naturally stacked ReSe2. How can we control the sliding on the specific layers? Or is it a random phenomenon in multilayers configuration?
4. Have the authors tried to achieve a deeper insight concerning the nanoscale electrical characterization, employing alternative techniques (like C-AFM, KPFM...) to make comparison with macroscale electrical characterizations?
5. Could the sliding affect locally also the band gap of these multilayers ReSe2? Have the authors tried evaluation of photoluminescence emission?
6. Have the authors some idea to improve the spontaneous polarization of this multilayers ReSe2 device?
7. Concerning the measurement of the endurance of ferroelectric switches, the device is graphene/ReSe2/BN/Au as described in the text, or BN/graphene/ReSe2/BN/Au as described in the supporting information?
8. Can these multilayers ReSe2 be comparable or better with respect to other 2D materials based sliding ferroelectric devices?

Reviewer #3

(Remarks to the Author)

Version 1:

Reviewer comments:

Reviewer #1

(Remarks to the Author)

I have reviewed the revised version of the manuscript "Emergence of sliding ferroelectricity in naturally parallel-stacked multilayer ReSe2 semiconductor". I appreciate the authors' comprehensive responses to the comments raised in the previous round of review.

The authors have adequately addressed all of the concerns I previously noted. The revised manuscript reflects significant improvements in clarity, methodology, and overall presentation. The manuscript is now well-structured, scientifically sound, and presents findings that are of interest to the field. I have no further major concerns.

Reviewer #2

(Remarks to the Author)

The authors have satisfactorily answered all questions.
In my opinion, the paper is now ready for publication.

Reviewer #3

(Remarks to the Author)

List of Responses and Revision

Title: “Emergence of sliding ferroelectricity in naturally parallel-stacked multilayer ReSe₂ semiconductor”

ALL THE CHANGES WERE DONE IN RED IN THE REVISED MANUSCRIPT.

REVIEWER COMMENTS

Reviewer #1 (Remarks to the Author):

General Comments:

I have read carefully the work “emergence of sliding ferroelectricity in naturally parallel stacked multilayer ReSe₂ semiconductor”. The paper reports the first observation of sliding ferroelectricity in ReSe₂ semiconductor with parallel stacking providing experimental and theoretical evidence. The paper is relevant as the field of sliding ferroelectricity is of wide interest in the community and the system presented does not require artificial stacking of different 2D materials or at a twisted angle. The authors first investigate the fundamental physics of the system from a theoretical perspective, then support their findings with direct experimental measurements using multiple well-established techniques. Moreover, they explore the potential integration of sliding ferroelectric ReSe₂ in functional devices. While there are already several Van der Waals systems which present sliding ferroelectricity without artificial stacking e.g. InSe:Y (Ref. 15) or CuInP₂S₆ (ref. 35), the supporting data is solid and I would still recommend the acceptance in Nature Communications with the revisions that I list in the following:

Response: We thank the reviewer for his/her positive comments on our work. We have carefully considered all the reviewers’ comments/suggestions and replied to the comments point by point as follows.

STYLE AND CLARITY

Q1. Line 80: I would advise against the use of “fantastic” as an adjective for quantifiable properties in scientific writing. I suggest removing it and leave the correct and formal

communication “the displacements of microscopic atomic configuration are directly visualized using scanning transmission electron microscopy (STEM)”

Response: We appreciate your meticulous review and valuable suggestion. We strongly agree with the reviewer's suggestion and remove the word “fantastic” in the revised manuscript.

Q2. It would be appreciated a general improvement of the captions, especially the ones of Supplementary that appear approximative and not well deepened and effective.

Response: We are very grateful to you for the valuable suggestions. Based on the reviewers' suggestions, we have further generally improved the captions in the manuscript. The corresponding corrections were added to the revised manuscript and Supplementary Information.

Q3. In Fig. 1, panels d-e-f might use the same color for the same thickness to ease readability.

Response: Thanks for your meticulous suggestion. We used the same color for the same thickness in panels d-e-f of Fig. 1 for readability (**Fig. R1**), and the corresponding changes were implemented in the revised manuscript (**Fig. 1 d-f**).

Fig. R1 Crystal structures and spectral characteristics of ReSe₂ flakes.

Q4. I would suggest the authors to improve the legend and color-scale of Fig. S11.

Response: Thanks for your meticulous suggestion. We sincerely apologize for the unclear legend and color-scale in the original figure. We improved the corresponding legend and color-scale (**Fig. R2**) and added them to the revised Supplementary Information (**Supplementary Fig. 13**).

Fig. R2 The switchable diode characteristics after applying different poling voltages.

ISSUES

Q1. Introduction: *The authors are right in pointing out that many sliding ferroelectrics based on Van der Waals materials require twisted or artificial stacking to achieve a net switchable polarization. The fact that ReSe₂ does not need artificial stacking is highlighted in the text so it should be the same for the other systems already known for this property. A proper comparison of the advancement with respect to Ref. 15 and Ref. 35 should be given in the introduction.*

Response: Thank you for your insightful comments that guided us towards a more comprehensive literature review in the introduction section. After carefully examining the Ref. 15 and Ref. 35, we realize that an important sliding ferroelectric material (Y-InSe) was not given in the introduction of our original manuscript. Y-InSe achieves enhanced sliding switchable polarization by modifying the microstructure of the InSe system through Y-doping strategy, and avoids the complex artificial stacking approach in sliding ferroelectrics, providing a new paradigm for obtaining easy-to-prepare and high-performance ferroelectric semiconductors. Our work, which also eliminates the need for meticulous artificial stacking and complex growth processes, reveals for the first time the sliding ferroelectricity of the intrinsic parallel-stacked ReSe₂ system and further explores its potential for ferroelectric photovoltaic applications. This dual exploration of rare earth element doping and intrinsic sliding ferroelectrics provides a feasible solution for realizing high-performance and easy-to-prepare sliding ferroelectrics. It should be clarified that we only compared the advancement of Ref. 15, because the ferroelectricity of Ref. 35 is not caused by interlayer sliding.

We have included this important material in the introduction section of the revised manuscript (Pages 3, 4).

Q2. Line 170: “Pt substrate has good smoothness”. Which is the surface roughness? Does it present significant ripples that can favor out of plane polarization? How are these substrates made? There is nothing in the methods section on this.

Response: Thanks for your important comments and questions. As shown in Fig. R3, we measured the roughness of the Pt substrate, and the roughness is about 0.29 nm. There is no ripple phenomenon, which will not affect the characterization of the OOP polarization. In addition, the Pt substrate we used in this work is commercially available and purchased from Hefei Kejing Material Technology Co., Ltd. We consulted the company and confirmed that Pt is grown by magnetron sputtering. We have added the source of the Pt substrate in the revised manuscript to avoid misunderstanding.

Fig. R3 The roughness of the Pt substrate.

Q3. Line 185: for Au/ReSe₂/Au devices thicknesses should be expressed, particularly the number of layers ReSe₂ is fundamental to compare the latter results.

Response: Thanks for your kind reminder and suggestion. The thickness of the ReSe₂ flake is ~15.5 nm (Fig. R4b), and the parameter was also added to the revised manuscript on Page 10 and Supplementary Information (Supplementary Fig. 12d).

Fig. R4 a An optical image of Au/ReSe₂/Au device. **b** Corresponding height profile of ReSe₂ flake.

Q4. Line 190: the difference between theoretical and experimental polarization is attributed to the number of sample and sliding layers: what are possible ratios that would justify this value? Is it possible that the lower value comes from the formation of domains, i.e. a partial switching of the device? If possible, a PFM map taken on a capacitor device would help in clarifying this point.

Response: Thank you very much for your professional suggestion. Regarding the analysis of the difference between theoretical and experimental polarization values, we fully agree with your perspective that the lower experimental values may originate from the partial switching phenomenon of ferroelectric domains. In addition, the number of sample layers and sliding layers may also be a key factor leading to the difference between theoretical and experimental polarization. The details are described as follows:

Effects of the number of sample layers and sliding layers

We have verified the polarization values of sliding different layers in a fixed layer sample and sliding only one layer in different layer samples through theoretical calculations. The results indicate that sliding different layers in six-layer sample will result in significantly different polarization values (**Fig. R5a**), and sliding only one layer in samples with different layers can also lead to different ferroelectric polarization values (**Fig. R5b**). In the experiments, due to the inability to determine the specific number of sliding layers in each sample, even in samples with the same number of layers, this can lead to differences in the ferroelectric polarization values. Therefore, both the number of sample layers and the specific number of sliding layers affect the ferroelectric polarization values.

Effect of partial switching of ferroelectric domains

Following the reviewer's suggestion, to verify the effect of partial polarization reversal on the experimental polarization values being smaller than theoretical values, we directly tested the distribution and dynamic evolution process of ferroelectric domains at different voltages by PFM phase imaging. **Fig. R6a, b** shows the topography and initial phase images of 14.7 nm ReSe₂ in capacitor device, clearly revealing the opposite ferroelectric domains. After applying 4 V and 6 V polarization voltages, the PFM phase changes significantly with increasing voltage (**Fig. R6c, d**). Notably, when the polarization voltage increases to 7 V, the ferroelectric domains hardly change and do not completely reverse (**Fig. R6e**). This phenomenon verifies the explanation for partial switching of ferroelectric domains.

Overall, the difference between theoretical and experimental polarization may be attributed to two factors: partial switching of ferroelectric domains and the differences in the number of sample and sliding layers. The related descriptions were added to the revised Manuscript on **Page 10**.

Fig. R5 a Ferroelectric polarization values of sliding different layers in a fixed six-layer system.

b Ferroelectric polarization values of sliding one layer in different layer systems.

Fig. R6 a-e Topography and PFM phase images of ferroelectric capacitor device under initial

state and after applying different poling voltages. Scale bar: 500 nm.

Q5. Line 198: Again, a better description of the device stack with layer thicknesses is advised.

Response: Thank you for your kind reminder. The thickness of the ReSe₂ flake is ~17.9 nm (Fig. R7b), and the corresponding description was also added to the revised manuscript on Page 11 and Supplementary Information (Supplementary Fig. 12e).

Fig. R7 **a** An optical image of BN/Gr/ReSe₂/BN/Au device. **b** Corresponding height profile of ReSe₂ flake.

Q6. Line 227: the claim that ReSe₂ photovoltaic devices are promising for optoelectronic devices at the current status is far-fetched and unsupported unless some quantitative performance advantage is provided.

Response: Thanks for your constructive comment and suggestion. For a more scientific statement, we have removed the relevant descriptions that ReSe₂ are promising for optoelectronic devices. In addition, we have tried to compare the relationship between open-circuit voltage (V_{oc}) and optical power for different 2D sliding ferroelectric systems. As shown in Fig. R8, compared to the widely reported 3R-MoS₂, 3R-WS₂, ϵ -InSe, and γ -InSe [Science Advances, 8, eade3759 (2022); Nature Communications, 16, 230 (2025); Advanced Materials, 37, 2416117 (2025); Advanced Materials, 36, 2410696 (2024)], larger V_{oc} values were achieved at smaller optical power densities in our ReSe₂ ferroelectric photovoltaic device. These results have also been added to the revised Supplementary Information (Supplementary Fig. 17) and the corresponding descriptions have also been added to the revised manuscript on Page 12.

Fig. R8 Comparison of V_{oc} of 1T' ReSe₂ ferroelectric photovoltaic device with that of other sliding ferroelectric photovoltaic devices.

Q7. Fig 5c: the pulse duration employed for the endurance test should be reported in the caption and in the main text.

Response: Thank you for your kind reminder. In the manuscript, the pulse duration of the durability test is 1 s. We also added the pulse duration of the durability test in the caption of **Fig.5c** and main text on **Page 11** of the revised manuscript.

OTHER QUESTIONS:

Q1. Concerning the power-dependent SHG, such exposition did lead to any degradation of the flake? Did the authors also perform temperature-dependent SHG experiments to assess the Curie temperature of the material?

Response: Thank you for your professional question and suggestion. To investigate the effects of laser power on 1T'-ReSe₂ flakes, we first performed Raman spectra on fresh samples (**blue line in Fig. R9a**). Subsequently, the SHG signals of 1T' ReSe₂ were measured at different powers (5 mW~100 mW) (**Fig. R9b**). Finally, we conducted Raman spectra measurement again, which showed no significant changes for the Raman peaks (**red line in Fig. R9a**). These results indicate that power-dependent SHG testing does not result in degradation of flake under the current experimental conditions.

In addition, we further investigated the temperature-dependent SHG intensity of ~15 nm 1T' ReSe₂ to determine its Curie temperature. **Fig. R9c** shows the normalized intensity plotted as a function of temperature. A clear SHG signal was observed for 1T' ReSe₂ at room temperature,

and the SHG intensity decreases with increasing temperature. The Curie temperature of 1T' ReSe₂ was estimated to be approximately 490 K. The relevant results were added to the revised Supplementary Information (**Supplementary Fig. 11**) and the corresponding descriptions were also added to the revised manuscript on **Page 10**.

Fig. R9 **a** Raman spectra of ~15 nm 1T' ReSe₂ flake in the fresh state and after SHG testing at different power levels. **b** Power-dependent SHG intensity of 1T' ReSe₂ flake under a 1064 nm laser. **c** Temperature-dependent of SHG intensity.

Q2. Regarding the PFM measurements, did the authors estimate the piezoelectric coefficient (i.e. piezoelectric modulus d_{33})?

Response: Thanks for your constructive question. We evaluated the piezoelectric coefficient d_{33} of the ReSe₂ flake (8.5 nm) exfoliated on a Pt/Ti/SiO₂/Si substrate using PFM. As shown in **Fig.R10**, when the driving AC voltage (V_{AC}) is scanned from 1 V to 5 V, the OOP amplitude signal of the sample significantly increases. By fitting the amplitude of the sample to V_{AC} , the effective d_{33} of the 1T' ReSe₂ flake was estimated to be approximately 3.83 pm V⁻¹. We also added the piezoelectric response results and corresponding descriptions to the revised Supplementary Information (**Supplementary Fig. 10**) and the revised manuscript on **Page 10**.

Fig. R10 **a** Height image and OOP intrinsic amplitude images of 1T' ReSe₂ flake under

different driving AC voltages. **b** Corresponding OOP amplitude evolution curve with different driving AC voltages.

Q3. Could the authors justify the smaller out-of-plane amplitude of the PFM hysteresis loop with 10 layers with respect to the one with 3 layers (figure 4c-f)? Did the authors verify how many sliding layers were present in each of the mentioned samples?

Response: Thanks for your meticulous comments and questions. The amplitude value of sliding ferroelectric system may depend on many factors, such as the magnitude of interlayer shear displacement, specific sliding layer numbers, etc., thus we cannot directly give a definite conclusion about the different out-of-plane amplitudes. We are very grateful for your constructive questions that can contribute to the development of the field, and that need to be further discussed and resolved in sliding ferroelectrics including ReSe₂. In the present study, we are focusing on demonstrating the emergence of sliding ferroelectricity in naturally parallel-stacked multilayer ReSe₂ semiconductor.

Additionally, we fully recognize the importance of determining the number of sliding layers in each tested sample. However, under our current experimental conditions, there are fundamental technical challenges to identify the number of sliding layers in each of the mentioned samples. Especially for thin sliding ferroelectric samples, TEM observation requires prior FIB (Focused Ion Beam) processing of the sample, which may damage the thin-layer structure and consequently compromise the accurate determination of sliding layer configurations.

Q4. Could the authors comment on the peculiar left-side of the butterfly hysteresis loop of Fig. 4f? Such branch does not seem to be open as common ones.

Response: Thank you for your in-depth attention to the phenomenon of crossover and lacking openness on the left side of the amplitude-electric field loop. This phenomenon was also observed in other reported works [Nature Communications, 15, 3799 (2024); Advanced Materials, 36, 2404177 (2024); Journal of the American Chemical Society, 142, 18592 (2020)]. It may originate from the following reasons: i) The enhanced screening effect in the ferroelectric semiconductors [Nature Communications, 16, 365 (2025); ACS Nano, 14, 7628 (2020)] exhibits asymmetry during forward and reverse scans, which may lead to the

phenomenon of crossover and lack of openness in amplitude hysteresis loops. ii) Defects such as vacancies in the material may lead to crossover phenomena of amplitude hysteresis loop through defect pinning effects, and also affect the openness of the amplitude hysteresis loop [Ferroelectrics, 413, 371-380 (2011); Materials Characterization, 176: 111131 (2021)]. iii) During the ramp process of PFM measurements, the voltage will first jump from 0 V to negative voltage and then gradually increase, which will cause the curve to be initially unstable. After continuous testing, it was found that the crossover phenomenon would disappear. We measured other points of the flake with continuous ramp, and no crossover phenomenon was observed in the amplitude hysteresis loop (**Fig. R11**). To avoid ambiguity, we replaced the PFM amplitude and phase hysteresis loops in **Fig. 4f**.

Fig. R11 PFM amplitude (blue) and phase (read) hysteresis loops of 3L ReSe₂ flake.

Thanks again for your professional comments, which greatly help us improve the manuscript. Hopefully, we have addressed all of your concerns.

Reviewer #2 (Remarks to the Author):

General comments:

Review on the paper entitled: “Emergence of sliding ferroelectricity in naturally parallel-1 stacked multilayer ReSe₂ semiconductor” by W. Xue et al. The authors proposed an alternative way compared to the state-of-the-art to achieve sliding ferroelectricity in two-dimensional ReSe₂ systems. By combining both theoretical and experimental results, the authors described clearly the observation of sliding ferroelectricity on multilayers stacked ReSe₂, with a number of layers ≥ 3 . The paper is well written and argued. From my point of view, some minor revisions are necessary before publication. My curiosities and suggestions are reported below:

Response: We thank the reviewer for his/her positive comments on our work. We have carefully considered all comments/suggestions of the reviewer. According to the reviewer’s valuable suggestions, the manuscript was revised accordingly as in the follow.

Q1. *A first suggestion concerns the description of material preparation. After the general ReSe₂ description in Figure 1, the authors passed to the experimental Raman description without information about the ReSe₂ flake preparation, which will be treated subsequently for the final device section. My suggestion is to anticipate the preparation description between the first two figures. This could also help the switching between the theoretical and experimental description during the reading.*

Response: We appreciate your meticulous review and valuable suggestions. The description of the material preparation has been added between Fig. 1a and 1b in the revised manuscript (**page 6**) for better readability. Specific preparation methods have been added in the “Methods” section on **page 13** of the revised manuscript. The specific preparation process of ReSe₂ flakes is as follows: Firstly, the ReSe₂ crystals were synthesized by chemical vapor transfer (CVT) method (provided by Nanjing MKNANO Tech. Co., Ltd.). Then, the ReSe₂ flakes with different thicknesses were mechanically exfoliated from the bulk crystals using polydimethylsiloxane (PDMS). Finally, the exfoliated flakes were transferred to the target substrates by the dry transfer technique and heated at 80~90° C for 10 minutes to release the adhesion of PDMS to ensure the successful transfer of the flakes.

Q2. *Have the authors observed some frequency-dependence on the number of ReSe₂ layers,*

like those observed in other 2D TMDs (such as MoS₂, WS₂...)?

Response: We appreciate your insightful comments and questions. 2D materials usually have frequency-dependent properties, such as frequency-dependent Raman in 2D TMDs. Therefore, we observed frequency-dependent Raman properties in ReSe₂ samples with different layer numbers, and the specific results are as follows:

Under conventional high-frequency Raman conditions (532 nm excitation), the Raman peak positions of ReSe₂ show negligible variation with increasing layer numbers (**Fig. R1a**), which is different from the strong thickness-dependent Raman spectra variation in most 2D layered materials and is attributed to the weak interlayer coupling [ACS nano, 10, 2752-2760 (2016)]. Therefore, the high-frequency Raman spectra of ReSe₂ cannot be utilized to identify their layer numbers as that used in other 2D materials.

The weak van der Waals coupling in 2D layered materials usually leads to the emergence of interlayer phonon modes which are typically located in ultralow-frequency region [ACS nano, 10, 2752-2760 (2016); Nano letters, 16, 1404-1409 (2016); Nanoscale, 8, 8324-8332 (2016)]. Hence, the ultralow frequency Raman spectra can offer valuable information about the interlayer charge exchanges, screenings, scatterings and their stacking order, rendering it a favorable sign for determining the layer number of the 2D material [Advanced Science, 7, 2002320 (2020)]. The interlayer shear and layer-breathing Raman modes of a layered material directly reflect its interlayer vdW coupling. As shown in **Fig. R1b**, these modes only occur in multilayer ReSe₂ and not in the monolayer. The positions of the Raman peaks change significantly as the sample layer number increases, as reported in the reference [Nano Research, 8, 3651-3661 (2015)]. Therefore, the low-frequency Raman spectra of ReSe₂ can be used to identify their layer numbers similar to other 2D TMDs.

Fig. R1 a, b High-frequency and ultralow frequency Raman spectra of monolayer and few-layer ReSe₂ flakes.

Q3. Another aspect that was not clarified (or not clear after my readings) is the reason of these sliding on naturally stacked ReSe₂. How can we control the sliding on the specific layers? Or is it a random phenomenon in multilayers configuration?

Response: Thanks for your professional questions. We sincerely apologize for not describing clearly the reason for the sliding on naturally stacked ReSe₂. The sliding reasons in ReSe₂ can be explained as follows:

Different from the most common group VI TMDs (such as MoS₂ and WS₂), there are unbound valence electrons outside the nucleus of the Re atom in ReSe₂. Due to the Peierls distortion, the adjacent Re atoms are bonded in the form of zigzag Re₄ clusters and align along the direction of the lattice vector *b* to form Re₄ chains, forming a distorted octahedral (1T') structure and reducing the symmetry of structures [Advanced Functional Materials, 33, 2212167 (2023)]. In addition, the Peierls distortion greatly weakens the van der Waals interaction in the interlayer. DFT calculations demonstrate that the interlayer coupling energy of ReS₂ is about 18 meV per unit cell, while that of MoS₂ is about 460 meV for the 2 × 2 conventional cells [Nature communications, 5, 3252 (2014)]. ReSe₂ also exhibits weak interlayer interactions similar to ReS₂. The low-symmetry structure and weak van der Waals

interlayer interactions of ReSe₂ endow it the unique ability of easy interlayer sliding, which facilitates the realization of low-potential-barrier interlayer sliding [Advanced Materials, 36, 2404734 (2024)].

Currently, in multilayer sliding ferroelectrics, it is usually difficult to control the sliding of specific layers. Since the sliding energy of each layer is equivalent, it is generally believed that sliding occurs in a certain layer of the multilayer system. Thanks again for your constructive and inspired question.

The related descriptions were added to the revised manuscript on **Page 5**.

***Q4.** Have the authors tried to achieve a deeper insight concerning the nanoscale electrical characterization, employing alternative techniques (like C-AFM, KPFM...) to make comparison with macroscale electrical characterizations?*

Response: Thanks for your insightful suggestion and question. According to the reviewer's suggestion, we conducted the related C-AFM and KPFM experiments to gain a deeper understanding of the nanoscale electrical characterization. First, the C-AFM behaviors of 15 nm thick 1T' ReSe₂ transferred onto a Pt substrate were measured. As shown in **Fig. R2**, the *I-V* loop was significantly open when a -6 V voltage bias was applied, indicating that the sample was polarized to the downward state. Then, three positive voltages of 3 V, 6 V, and 7 V were applied. The results show that the current of the device increases at a 3 V bias and an obvious open loop was formed applying a 6 V bias with the polarization switching to the upward, and the *I-V* loop was not open at a 7 V bias because the upward polarization state has already occurred. This phenomenon is consistent with macroscopic electrical behavior, that is, the partial ferroelectric domains were reversed at a smaller voltage, while a larger bias can provide sufficient driving force to switch more domains. In addition, KPFM measurements of ReSe₂ flake were conducted after applying positive 3 V and 6 V polarization voltages (**Fig. R3**). The results reveal that the surface potential of the polarized region is significantly increased compared to the unpolarized region, and the potential difference reaches 40.7 mV (3 V) and 65.9 mV (6 V), respectively. A higher polarized electric field can more sufficiently drive ferroelectric polarization reversal, resulting in generating a larger potential difference. The voltage amplitude dependence of the ferroelectric polarization exhibited in the above nanoscale

electrical properties is consistent with the macroscale electrical properties.

These results were added to the revised Supplementary Information (**Supplementary Fig. 15 and Fig. 16**) and the corresponding descriptions have also been added to the revised manuscript on **Page 11**.

Fig. R2 C-AFM I - V curves of Pt/1T' ReSe₂/Pt device at different DC bias voltages.

Fig. R3 **a** Surface potential images of 1T' ReSe₂ after +3 V and +6 V poling voltages. The yellow dashed boxes indicate the area of applied voltage. **b** The change of surface potential acquired from the red dashed line in (a).

Q5. *Could the sliding affect locally also the band gap of these multilayers ReSe₂? Have the authors tried evaluation of photoluminescence emission?*

Response: Thanks for your valuable questions and suggestions. We tried to understand the effect of the sliding on the bandgap of multilayer ReSe₂ from both theoretical and experimental perspectives.

According to the theoretical and STEM results, ReSe₂ exhibits the interlayer sliding-dependent polarization state, so we further explore the effect of sliding on its electronic structure. The

electronic band structure and density of states (DOS) indicate that the band gap of parallel-stacked ReSe₂ is 1.15 eV (**Fig.R4a, d**). The bandgap reduces to 0.97 eV after switching from parallel stacked state to polar state by interlayer sliding (**Fig.R4b, c**), confirming that interlayer sliding affects the bandgap. Based on the obtained electronic structures, we further calculate the frequency-dependent dielectric function (ϵ_2) and optical absorption coefficient (α) using the independent particle approximation implemented [Physical Review B, 73, 045112 (2006)], as shown in **Fig. R5a and R5b**. The calculated imaginary dielectric function and absorption coefficient determine the optical absorption properties of ReSe₂. It indicates that optical absorption coefficient of the polar state is larger than those of the parallel stacking within the visible photon energy range.

We also further investigate the effect of sliding on band gap of 1T' ReSe₂ (15 nm) by PL spectroscopy. As shown in **Fig. R5c**, the PL peak appears initially near 1.29 eV for fresh ReSe₂. After applying a 6 V poling voltage, the PL peak occurs red shift (1.24 eV) and intensity weakens. The trend of our experimental results is consistent with theoretical calculations, both indicating that the interlayer sliding induces band gap narrowing.

Fig. R4 a-d Electronic band structure and density of states for parallel stacked, intermediate, and polar states.

Fig. R5 a, b Dielectric function ϵ_2 and absorption coefficient α for parallel stacking, intermediate and polar states. **c** PL spectrum of 1T' ReSe₂ flake at initial and after applying 6 V poling voltage.

Q6. Have the authors some idea to improve the spontaneous polarization of this multilayers ReSe₂ device?

Response: Thanks for your professional and insightful question. It is great important for improving the spontaneous polarization of the sliding ferroelectrics. Doping, especially with rare-earth elements, has been proven to be an effective method for improving the spontaneous polarization of ferroelectrics, which is due to it can modulate the lattice structure (e.g., bond distances and angles, atomic positions) through microstrain/structural constraints. For example, the orthorhombic phase (Pca2₁) of HfO₂-based ferroelectric materials faces the challenge of phase stability due to its metastable properties. The conventional approaches utilize the lower surface energy of the orthorhombic phase to stabilize the ferroelectric phase by reducing the grain size. However, this method introduces numerous defects, resulting in the ferroelectric polarization far below theoretical value. Yun et al. [Nature Materials, 21, 903-909 (2022)] prepared Y-doped HfO₂ thin films with small rhombohedral distortion, in which Y-doping promotes the stability of the ferroelectric orthorhombic phase and suppresses the formation of non-ferroelectric phases by inducing lattice distortion and charge compensation. The observed ferroelectric polarization is close to the theoretical prediction. In the ferroelectric InSe system, many natural stacking-faults are observed due to very low formation energy, which is detrimental to the long-range ordering and thermodynamic stability of the sliding ferroelectricity. Wang et al. [Nature Communications, 14, 36 (2023)] introduced anisotropic stress/strain through Y-doping to control the microstructure of InSe, including the elimination

of stacking-faults and a subtle rhombohedral distortion, thus enhancing the spontaneous polarization of InSe.

For the ReX_2 systems, numerous studies have been conducted on modulating the lattice structure of ReX_2 through cation doping strategies [Advanced Science, 7, 2002320 (2020); Advanced Materials, 34, 2202722 (2022)], Therefore, doping is likely to be an important means of improving the spontaneous polarization of ReSe_2 .

Q7. Concerning the measurement of the endurance of ferroelectric switches, the device is graphene/ReSe₂/BN/Au as described in the text, or BN/graphene/ReSe₂/BN/Au as described in the supporting information?

Response: Thank you for the kind reminder. We apologize for the ambiguity of the device structure in the manuscript. Regarding the measurement of ferroelectric switch durability, our device structure is BN/graphene/ReSe₂/BN/Au, where the upper BN layer here acts as an encapsulation layer to prevent the contact of ReSe₂ with air. The relevant revisions were added to the revised manuscript on **Page 11**.

Q8. Can these multilayers ReSe₂ be comparable or better with respect to other 2D materials based sliding ferroelectric devices?

Response: Thanks for your insightful question. In our manuscript, the photovoltaic performance of 1T' ReSe₂ ferroelectric device was demonstrated. Therefore, we compared the relationship between open-circuit voltage (V_{oc}) and optical power for different 2D sliding ferroelectric systems. As shown in **Fig. R6**, compared to the widely reported 3R-MoS₂, 3R-WS₂, ϵ -InSe, and γ -InSe [Science Advances, 8, eade3759 (2022); Nature Communications, 16, 230 (2025); Advanced Materials, 37, 2416117 (2025); Advanced Materials, 36, 2410696 (2024)], larger V_{oc} values were achieved at smaller optical power densities in our ReSe₂ ferroelectric photovoltaic device. These results have also been added to the revised supplementary information (**Supplementary Fig. 17**) and the corresponding descriptions have also been added to the revised manuscript on **Page 12**.

Fig. R6 Comparison of V_{oc} of our 1T' ReSe₂ ferroelectric photovoltaic device with that of other photovoltaic devices with sliding ferroelectric materials.

Thanks again for your professional comments, which greatly help us improve the manuscript. Hopefully, we have addressed all of your concerns.

List of Responses and Revisions

Title: “Emergence of sliding ferroelectricity in naturally parallel-stacked multilayer ReSe₂ semiconductor”

Authors: Wuhong Xue, Peng Wang, Wenjuan Ci, Ying Guo, Jingyuan Qu, Zeting Zeng, Tianqi Liu, Ri He, Shaobo Cheng, Xiaohong Xu

ALL THE CHANGES WERE DONE IN RED IN THE REVISED MANUSCRIPT.

Response to Reviewer 1:

Comments:

I have reviewed the revised version of the manuscript "Emergence of sliding ferroelectricity in naturally parallel-stacked multilayer ReSe₂ semiconductor". I appreciate the authors' comprehensive responses to the comments raised in the previous round of review.

The authors have adequately addressed all of the concerns I previously noted. The revised manuscript reflects significant improvements in clarity, methodology, and overall presentation. The manuscript is now well-structured, scientifically sound, and presents findings that are of interest to the field. I have no further major concerns.

Response: Thank you very much for your positive feedback on our research. Your constructive suggestions have been invaluable in improving the quality of our work.

Response to Reviewer 2:

Comments:

The authors have satisfactorily answered all questions. In my opinion, the paper is now ready for publication.

Response: Thank you very much for your positive comments. Your valuable feedback has significantly contributed to the improvement of our work.

Response to Reviewer 3:

Comments:

Response: Thank you very much for your participation in the peer review process. We appreciate the collaborative effort in reviewing our manuscript and the valuable insights provided.